# Design, Synthesis, and Biological Evaluation of Novel MAO-A Inhibitors Targeting Lung Cancer

**DOI:** 10.3390/molecules27092887

**Published:** 2022-04-30

**Authors:** Sanaa Bardaweel, Reem Aljanabi, Dima Sabbah, Kamal Sweidan

**Affiliations:** 1Department of Pharmaceutical Sciences, School of Pharmacy, The University of Jordan, Amman 11942, Jordan; aljanabireem@live.com; 2Department of Pharmacy, Faculty of Pharmacy, Al-Zaytoonah University of Jordan, P.O. Box 130, Amman 11733, Jordan; dima.sabbah@zuj.edu.jo; 3Department of Chemistry, The University of Jordan, Amman 11942, Jordan; k.sweidan@ju.edu.jo

**Keywords:** MAO-A inhibitors, anticancer, lung cancer, docking, pharmacophore, NCI, virtual screening

## Abstract

Lung cancer is one of the most common causes of cancer-related deaths worldwide. Monoamine Oxidase-A (MAO-A) enzyme mediates the production of reactive oxygen species (ROS) that trigger DNA damage and oxidative injury of cells resulting in tumor initiation and progression. Available MAO-A inhibitors are used as antidepressants, however, their role as anticancer agents is still under investigation. Ligand- and structure-based drug design approaches guided the discovery and development of novel MAO-A inhibitors. A series of 1H indole-2-carboxamide derivatives was prepared and characterized using 1H-NMR, 13C-NMR, and IR. The antiproliferative effects of MAO-A inhibitors were evaluated using the cell viability assay (MTT), and MAO-A activity was evaluated using MAO-A activity assay. The presumed inhibitors significantly inhibited the growth of lung cell lines in a dose- and time dependent manner. The half maximal inhibitory concentration (IC_50_) values of MAO-A inhibitors (**S1, S2, S4, S7,** and **S10**) were 33.37, 146.1, 208.99, 307.7, and 147.2 µM, respectively, in A549. Glide docking against MAO-A showed that the derivatives accommodate MAO-A binding cleft and engage with key binding residues. MAO-A inhibitors provide significant and consistent evidence on MAO-A activity in lung cancer and present a potential target for the development of new chemotherapeutic agents.

## 1. Introduction

Since its discovery in 1928, MAO enzyme has opened the door for the investigation of its effects in many biological processes [1]. Tyramine oxidase was the first discovered MAO enzyme. MAO enzyme is basically classified in two types: MAO-A and MAO-B. In general, MAO enzymes are known as groups of mitochondrial bound flavor proteins [2]. They work principally through catalyzing the oxidative deamination by converting monoamine into their corresponding aldehyde and ketone [3]. MAO plays a major role in controlling the level of many neurotransmitters including serotonin, dopamine, norepinephrine, epinephrine, tryptamine, tyramine, and 2-phenylethylamine [2,4]. Recently, studies have revealed a significant contribution of MAO (especially MAO-A) in the development and progression of carcinogenesis [5]. The effect is mainly related to MAO-A mediation in DNA damage through ROS production [6]. Advanced prostate cancer showed high expression of MAO-A that responded well to treatment with MAO-A inhibitor (clorgyline) [7,8]. Remarkably, prostate and renal cell carcinoma showed high expression of MAO-A [9]. Additionally, MAO-A enzyme showed a relatively high expression in bladder cancer and glioma [9]. Pargyline and phenelzine showed significant decrease in the neuroendocrine differentiation in prostate cancer. This effect occurs as a result of down-regulation of the repressor element-1 silencing transcription factor (REST) and activation of autophagy that is attributed to MAO inhibition [10]. Clorgyline resulted in a decrease in Temzolomide resistance, which is considered an essential problem in glioma treatment [11].

About 2.3 million cases of lung cancer were diagnosed in 2020, and 2 million deaths were expected as a consequence of such an increase in lung cancer rates in the same year [12]. Such an increasing rate urges researchers to focus efforts on finding new drugs targeting lung cancer. Lung cancer is usually diagnosed at an advanced stage where the disease has become incurable [13]. There are different types of available treatments ranging from radiotherapy to chemotherapy. Cisplatin and carboplatin, in addition to docetaxel, irinotecan, and topotecan, are considered the cornerstone for the treatment of non-small cell lung carcinoma (NSCLC) [14]. Interestingly, MAO-A showed high expression on both mRNA and protein level in NSCLC [15]. It is thought that MAO-A works primarily by affecting the epithelial to mesenchymal transition (EMT) state; that is considered a key step in the invasion and metastasis of cancer [16].

Molecular docking was first developed in the 1980s. Since that time, it became an important approach in the drug discovery process. Molecular docking works principally through molecular interaction prediction, where the protein and ligand are held together in a bound state. Docking relates mainly to structure-based drug design approach. The crystal structure of MAO-A/clogyline complex (PDB ID: 2BXS) was first released in 2005 [17,18]. The first generation of MAO inhibitors show many adverse effects, especially liver toxicity [19]. The cheese effect or the hypertensive crisis is also associated with the use of MAO inhibitors [19]. Therefore, efforts have focused on designing new MAO inhibitors with better activity and fewer side effects. Our study aims to provide a greater depth of understanding the role of MAO-A in lung cancer, exploring the binding interactions of MAO-A with known inhibitors, building a pharmacophore model for MAO-A inhibitors, and designing MAO-A inhibitors using structure-based and ligand-based approaches. Moreover, synthesizing novel MAO-A inhibitors, evaluating their anticancer activities against different lung cancer cell lines, and, finally, characterizing the activity of the synthesized MAO-A inhibitors against MAO isoforms will be explored.

## 2. Results and Discussion

### 2.1. Pharmacophore Modeling

We adopted the coordinates of ligand **MLG (S1)** in MAO-A (PDB ID: 2BXS) to build the 3D-sturctures of the reported inhibitors (**J1**-**J3**, **J5**, **J7**, **J10**, **J12**-**J13**, and **J26**) [20,21,22,23,24] (Figure 1). Superposing the modeled coordinates against **MLG** derived a pharmacophore model with the following features. F1: one aromatic or hydrophobic functionality, F2: one hydrophobic moiety, F3: one aromatic or hydrophobic motif, and F4: H-bond donor or acceptor. Our model indicates that MOA-A inhibitor should harbor two aromatic rings and one hydrophobic motif or three hydrophobic groups and one H-bond acceptor or donor moiety.

### 2.2. Pharmacophore Searching

We screened the generated pharmacophore model against the national cancer institute (NCI) database [25], harboring 265,240 compounds. Filtration of the NCI database was applied using Lipinski’s rule of five [26] to retrieve drug-like molecules. Lipinski’s rule of five describes the drug-likeness for oral active drugs, and it states oral drugs should have no more than 5 H-bond donors, no more than 10 H-bond acceptors, molecular weight equal to/or less than 500 Dalton, and partition coefficient (log P *o/w*) equal to/or less than 5. Fortunately, 52,457 of molecules were obtained and identified as hits. Further filtration against 52,457 hits retrieved 49 hits endorsing the MAO-A fingerprint (Figure 2). **S1** and retrieved hits (**S2**–**S10**) were provided by the NCI agency.

### 2.3. Molecular Docking

In order to discuss the anticancer activity of potential hits as MAO-A inhibitors (**S1**, **S2–S6**, **S7**, **S8–S10**) (Figure 3) [20,21,22,23,24,28] in human adenocarcinoma alveolar basal epithelial cell (A549), human non-small cell lung cancer cell (H1299), and human lung carcinoma cell (H661) lines, we employed the coordinates of MAO-A (PDB ID: 2Z5X) [29] to explore the structural basis of MAO-A/ligand complex formation. The complex of MAO-A/ligands was examined using Glide docking [30,31,32] studies against 2Z5X coordinates.

The Glide (grid-based ligand docking with energetics) is a comprehensive algorithm for the ligand/receptor docking strategy [30]. It provides high speed and accuracy levels for high-throughput virtual screening of millions of compounds along with superior accurate binding mode predictions [30]. Furthermore, the Glide score has been implemented with “extra precision” (XP) scoring function [31]. The XP Glide scoring protocol probes water desolvation energy and receptor–ligand structural backbones that are involved in complex formation and, consequently, enhance binding affinity [31].

The binding site of 2Z5X encloses Tyr69, Leu97, Ile180, Phe208, Ser209, Glu216, Cys323, Ile335, Thr336, Leu337, Met350, Phe352, Tyr407, and Tyr444. The binding sites accommodate hydrophobic and polar residues. The acidic and basic residues furnish an ionic (electrostatic) bonding, whereas the hydrophilic residues offer hydrogen-bonding, ion–dipole, and dipole–dipole interactions. The aromatic and hydrophobic residues provide π-stacking and van der Waals interactions, respectively. Docking data showed that the **S** series (**S1–S10**) accommodates the MAO-A binding domain (Figure 4A). Definitely, the docked pose of **J14** superposes the template of clorgyline (**S1**) in 2Z5X (Figure 4B). Results show that compounds form H-bond with Ala68, Tyr69, Asn181, Phe208, Gln215, and Gly443 of 2Z5X (Table 1). Interestingly, reported computational and experimental studies pinpointed the significance of these amino acids as key binding residues for MAO-A/ligand interaction [29,33,34,35,36,37,38,39].

To evaluate the execution of Glide algorithm, we compared the docked pose of F2M in MOA-B (PDB ID: 5MRL) [40] to its original orientation in the crystal assembly. Figure 5 demonstrates the superposing of the Glide-generated F2M conformation and its native geometry in 5MRL. The RMSD for heavy atoms of F2M between Glide-extracted docked conformation and the native pose was 1.048 Å. Result infers that Glide docking can successfully identify the native pose in crystal coordinates and, consequently, predict the ligand binding pose.

### 2.4. Biological Evaluation

The MTT assay was performed after exposing A549, H1299, and H661 cell lines to increasing concentrations of the presumed inhibitors for either 24, 48, or 72 h. Compounds **S1**, **S2**, **S4**, and **S10** have shown considerable antiproliferative activities, and the treated cells had reduced cell viability compared to the untreated control cells (Table 2). Specifically, **S2** exhibited potent antiproliferative activities against A549, H661, and H1299 with IC_50_ values of 33.37 µM, 10.76 µM, and 60.32 µM, respectively. In addition, **S4** demonestrated antiproliferative activities against A549, H661, and H1299 with IC_50_ values of 146.1 µM, 63.96 µM, and 65.46 µM, respectively. On the other hand, **S1** had significant antiproliferative activities against A549, H661, and H1299 with IC_50_ values of 208.99 µM, 215 µM, and 323.6 µM, respectively. Among the synthesized compounds, **S10** displayed noticeable antiproliferative activities against A549, H661, and H1299 with IC_50_ values of 147.2 µM, 73.83 µM, and 116 µM, respectively. Interestingly, compound **S7** showed selective antiproliferative activities against A549, H661 cell lines with IC_50_ values of 255.9 µM, and 168.8 µM, respectively, after treatment for 72 h. Moreover, compounds **S8** and **S9** showed preferential antiproliferative activities against H661. On the other hand, compounds **S3**, **S5**, and **S6** showed poor antiproliferative activities against all examined cell lines (Appendix A). To assess compound’s toxicity on normal cells, the MTT assay was performed after treatment of dermal fibroblast cells to increasing concentrations of MAO-A inhibitors for either 24, 48, or 72 h. Interestingly, **S1**, **S2**, **S4**, and **S10** did not show any antiproliferative activity on treated cells within the examined concentration range. Based on the estimated IC_50_ values, compounds **S1**, **S2**, and **S4** appear to have at least a three-fold difference in IC_50_ against fibroblast cells relevant to their IC_50_ against lung cancer cells. Compounds **S7** and **S10** showed no killing effect against normal cells. IC_50_ values against fibroblasts are shown in Appendix A.

In 2020, the world health organization announced that cancer was the leading cause of death in about 10 million persons all over the world. According to new cases in 2020, lung cancer was diagnosed in about 2.21 million cases [12]. Lung cancer was the most common cancer in men and the second most common cancer in women, as breast cancer was the most frequent cancer in women by 2.26 million cases [13,41]. Many difficulties have been associated with lung cancer treatment, ranging from side effects of chemotherapies, resistance to treatment, and high cost of radio and chemo therapies, which have urged the search for new therapeutic approaches to treat lung cancer. Many studies have shown the relationship between MAO-A enzyme and cancer. As MAO-A has been shown at increased levels in lung cancer, hence, the study of the MAO-A effect has become of great interest in lung cancer [42]. Non-small lung cancer has been associated with high expression of MAO-A. It was suggested that MAO-A leads to epithelial-mesenchymal transition (EMT) and the appearance of clinicopathological features and lymph node metastasis in late stage cancer [15]. Many non-selective MAO inhibitors have been tested on lung cancer cells. However, limited data are available for the use of MAO-A inhibitors in lung cancer treatment [43,44]. The current study was based on the previously reported antiproliferative activity of clorgyline (**S1**), a standard MAO-A inhibitor, against prostate and Glioma cell lines [8,10,11]. Clorgyline (**S1**) was considered as the reference compound for our study, so MAO-A inhibitors were chosen based on the pharmacophoric features and docking results. The present study has shown that MAO-A inhibitors (**S1**, **S2**, **S4**, **S7**, and **S10**) significantly inhibited the proliferation of A549, H1299, and H661 cells compared to the untreated control cells in a concentration and time dependent manner accompanied with a reasonable safety margin. Zhao et al. found that clorgyline (MAO-A inhibitor) has a significant antiproliferative effect against prostate cancer, even in the advanced stage [7]. The effect of MAO-A inhibitors on MAO-A activity in the lung cancer cell line was evaluated and an MAO-A activity assay was performed. Treatment of A549 with either **S1**, **S2**, **S4**, **S7**, or **S10** resulted in significant inhibition of MAO-A activity in comparison to the untreated control and the positive control, as shown in Table 3. MAO-A activity was measured using fluorometric detection (Ex/Em = 535 nm/587 nm), as illustrated in Figure 6.

It is well known that MAO-A is a mitochondrial enzyme which works through catalyzing oxidative deamination of neurotransmitters or dietary amines yielding H_2_O_2_. Studies on MAO-A activity in prostate cancer showed that clorgyline inhibited MAO-A activity and caused suppression in tumorigenesis and metastasis [10]. Pei-Chuan Li et al., 2015, showed that MAO-A expression and activity was higher in Hodgkin lymphoma (HL)-derived L1236, and usage of MAO-A inhibitor (clorgyline) revealed inhibition of cancer cells proliferation, invasion, and MAO-A activity [45]. Our study was the first study to verify the effect of MAO-A inhibitors on MAO-A activity in lung cancer cells. All MAO-A inhibitors showed activity inhibition in a dose dependent manner. These results were consistent with the docking results that demonstrated considerable binding scores against MAO-A.

## 3. Materials and Methods

### 3.1. Computational Methods

#### 3.1.1. Pharmacophore Generation

Three-dimensional structures of MAO-A inhibitors with diverse scaffolds: pyridazinylacetic acid derivatives [20], nicotinamides [21], indole alkaloids [21], flavonoids [21], xanthones [21,23], and anilides [24], were built and energy minimized using MMFF94X force field in MOE [23] software based on MLG template in MAO-A (PDB ID: 2Z5X) [29]. Each minimized structure was superposed to MLG (the most active molecule i.e., the lowest IC_50_ value). The superposed molecules were employed to define the pharmacophore model using the Pharmacophore Query module in MOE [23].

#### 3.1.2. Pharmacophore Search

The pharmacophore model was used to search against the National Cancer Institute (NCI) database to retrieve hit molecules. The pharmacophore search protocol was performed using MOE software [23].

#### 3.1.3. Preparation of Protein Structures

The X-ray coordinates of MAO-A (PDB ID: 2BXS) [39] and (PDB ID: 2Z5X) [29] with resolution 3.15 Å and 2.2 Å, respectively, were retrieved from the RCSB Protein Data Bank. We adopted the assembly of 2Z5X without ligand and transferred the ligand’s template (clorgyline: MLG) of 2BXS to 2Z5X; MLG was identified as a ligand to extract the grid file. The coordinates of 2Z5X were energetically minimized to avoid steric clash. Extra treatment for the minimized coordinates was executed using Protein Preparation algorithm in Schrödinger software [32] to maximize H-bond interactions between side chains.

#### 3.1.4. Preparation of Ligand Structures

The verified compounds (ligands) were modeled using MLG template in 2Z5X [29]. The ligands were built by MAESTRO [32] Build algorithm and energetically minimized by the MacroModel [32] script using OPLS2005 force field.

#### 3.1.5. Glide Docking

The Grid files for 2Z5X binding pocket were generated using the Glide Grid Generation panel with the bound ligands as centroid. During the docking process, the scaling factor for van der Waals for the nonpolar atoms was calibrated to 0.8 to provide some plasticity for the protein side chains. All other parameters were adjusted as defaults. The binding affinity was defined as docking score (Kcal/mol). The more negative the docking score, the better the binder.

### 3.2. Chemistry

#### 3.2.1. Chemicals

The following chemicals were purchased and used without further purification: indole-2-carboxylic acid 98% (Aldrich, Wyoming, IL, USA), oxalyl chloride 98% (Aldrich, Wyoming, IL, USA), 4-aminoacetophenone 97% (Aldrich, Wyoming, IL, USA), 3-aminoacetophenone 97% (Aldrich, Wyoming, IL, USA), *N*,*N*-dimethylformamide (DMF) (HPLC grade Tedia, Fairfield, OH, USA), pyridine (Cerritos, CA, USA), sodium bicarbonate (NaHCO_3_) (Labachemie, Mumbai, India), sodium borohydride (NaBH_4_) (Aldrich, Wyoming, IL, USA), dichloromethane (HPLC grade Carbon group, Lincoln, UK), hexane (HPLC grade Tedia, Fairfield, OH, USA), ethyl acetate (AZ chem, Dongguan, China), tetrahydro furan (THF) (Honeywell, Offenbach am Main, Germany), acetone (AZ Chem, Dongguan, China), absolute ethanol (Honeywell, Offenbach am Main, Germany), chloroform (HPLC grade) (Honeywell, Offenbach am Main, Germany), and methanol (HPLC grade Biosolve, Dieuze, France).

#### 3.2.2. Hits

Based on ligand- and structure-based drug design approaches, the following hits were purchased from Sigma Aldrich, Darmstadt, Germany, as shown in Table 4.

#### 3.2.3. Synthesis of Target Compounds

The following compounds were synthesized based on reported chemical procedure [17]. The synthetic procedure has been optimized as shown under the relevant section. All relevant data of NMR and IR were provided in the Appendix A.

All synthesized products were characterized by ^1^H NMR, ^13^C NMR on a Bruker-Avance III 500 MHz spectrophotometer in deutrated dimethyl sulfoxide (DMSO-*d*_6_) or deutrated methanol as solvents and TMS as internal standard. Chemical shift (*δ*) is expressed in ppm; coupling constant (*J*) values are given in Hertz (Hz). Infrared spectra were recorded using Shimadzu IR Affinity H spectrophotometer; all samples were prepared with potassium bromide and pressed into a disc. A Euro-Vector 8910 elemental analyzer-Weaver^TM^ software (Pavia, Italy) was employed.

The purity of all prepared compounds was examined using elemental analysis, and the results lay in the accepted limits (±0.5%) according to the instrument’s supplier. A standard material is usually analyzed prior and during the sample’s measurement to ensure the precision and accuracy of the analysis.

H-indole-2-carbonyl Chlorides: Adding oxalyl chloride (4.13 mmol) to a solution of 1H-indole-2-carboxylic acid (2.48 mmol) in 15 mL DCM at 0 °C followed by few drops of DMF. The solution was stirred for 30 min in an iced water bath, then refluxed for a further two hours. After that, the solution was cooled and evaporated under reduced pressure and the residue was used for the next step without further purification.

*N*-(4-acetylphenyl)-1H-indole-2-carboxamide (**S7**): A solution of 1H-indole-2-carbonyl chloride was prepared in 15 mL DCM in iced water bath, followed by addition of 4-aminoacetophenone solution (4.96 mmol). After waiting for 10 min, dry pyridine (1.5 mL) was added. The resulting solution was stirred overnight. The desired precipitate was filtered off and washed with DCM and then with 0.1 M NaHCO_3_ solution, and was dried for 20 min in an oven.

Pink solid, yield 79%, m.p. 290–293 °C. ^1^H-NMR (500 MHz, DMSO-*d6*) δ (ppm): 2.63 (s, 3H, CH_3_); 7.05 (t, *J* = 7.5 Hz, 1H, CH); 7.20 (t, *J* = 7.8 Hz, 1H, CH); 7.44–7.46 (m, 2H); 7.64 (d, *J* = 8.1 Hz, 1H); 7.89–7.99 (m, 4H), 10.49 (s, 1H, N-H amide); 11.79 (s, 1H, N-H indole). ^13^C-NMR (125 MHz, DMSO-d6) δ (ppm): 26.9 (CH_3_); 105.2 (C8); 112.9 (C3); 119.7 (C2’, C6’); 120.5 (C5); 122.4 (C6); 124.5 (C7); 127.4 (C4); 129.9 (C3’, C5’); 131.5 (C4’); 132.3 (C2); 137.5 (C9); 143.9 (C1’); 160.5 (C = O amide); 197.0 (C = O ketone). FT-IR (KBr-disc) cm^−1^: 3350 N-H amide; 3317 N-H indole; 1658 C = O for ketone and C = O amide. Elemental analysis for C_17_H_14_N_2_O_2_: calcd. C 73.37, H 5.07, N 10.07; found C 72.90, H 5.49, N 9.59.

*N*-(4-(1-hydroxyethyl)phenyl)-1H-indole-2-carboxamide (**S8**): (0.76 mmol) of sodium borohydride (NaBH_4_) was added to a solution of *N*-(4-acetylphenyl)-1H-indole-2-carboxamide (**S7**) in 10 mL of THF. The reaction mixture was stirred overnight at room temperature, then the solvent was evaporated under reduced pressure and the desired precipitate was collected and dried.

Beige solid, yield 90%, m.p. 253–255 °C. ^1^H-NMR (500 MHz, CD_3_OD-*d4*) δ (ppm): 1.42 (d, *J* = 6.0 Hz, 3H); 3.27 (m, 1H); 4.88 (d, *J* = 7.4 Hz, 2H); 7.05 (t, *J* = 7.1 Hz, 1H); 7.21 (t, *J* = 7.5 Hz, 1H); 7.28 (s, 1H); 7.33 (d, *J* = 7.7 Hz, 2H); 7.45 (d, *J* = 8.0 Hz, 1H); 7.61 (d, *J* = 7.7 Hz, 1H); 7.67 (d, *J* = 7.8 Hz, 2H), 7.91 (s, 1H, N-H amide); 7.99 (s, 1H, N-H indole). ^13^C-NMR (125 MHz, CD_3_OD-*d4*) δ (ppm): 24.2 (CH_3_), 69.1 (C-OH); 103.9 (C8); 111.7 (C3); 119.8 (C2’, C6’); 120.6 (C6); 121.5 (C5); 123.9 (C7); 125.5 (C3’, C5’); 127.7 (C4); 131.0 (C1’); 137.2 (C2); 137.3 (C9); 142.2 (C4’); 160.9 (C = O) amide). FT-IR (KBr-disc) cm^−1^: 3506 N-H amide; 3375 N-H indole; 3296 OH; 1649 C = O amide. Elemental analysis for C_17_H_16_N_2_O_2_: calcd. C 72.84, H 5.75, N 9.99; found C 72.41, H 6.17, N 9.63.

*N*-(3-acetylphenyl)-1H-indole-2-carboxamide (**S9**): A solution of 1H-indole-2-carbonyl chloride was prepared in 15 mL DCM in an iced water bath, followed by adding 3-aminoacetophenone solution (4.96 mmol). After 10 min, dry pyridine (1.5 mL) was added. The resulting solution was stirred overnight. The desired precipitate was filtered off and washed with DCM and then with 0.1 M NaHCO_3_ solution, and was dried for 20 min in oven.

White solid, yield 81%, m.p. 240–243 °C. ^1^H-NMR (500 MHz, DMSO-*d6*) δ (ppm): 2.45 (s, 3H, CH_3_); 7.04 (t, *J* = 6.7 Hz, 1H); 7.19 (t, *J* = 7.1 Hz, 1H); 7.43–7.47 (m, 2H), 7.50 (t, *J* = 7.4 Hz, 1H); 7.68 (q, *J* = 7.7 Hz, 2H); 8.11 (d, *J* = 7.4 Hz, 1H); 8.34 (s, 1H); 10.38 (s, 1H, N-H amide); 11.73 (s, 1H, N-H indole). ^13^C-NMR (125 MHz, DMSO-*d6*) δ (ppm): 27.2 (CH_3_); 104.6 (C8); 112.9 (C3); 119.8 (C2’); 120.5 (C6); 122.3 (C5); 123.9 (C7); 124.4 (C4’); 124.9 (C6’); 127.5 (C4); 129.6 (C5’); 131.7 (C3’); 137.4 (C2’); 137.8 (C9); 139.9 (C1’); 160.4 (C = O) amide; 198.2 (C = O) ketone. FT-IR (KBr-disc) cm^−1^: 3365 N-H amide; 3284 N-H indole; 1678 C = O for ketone and C = O amide. Elemental analysis for C_17_H_14_N_2_O_2_: calcd. C 73.37, H 5.07, N 10.07; found C 72.94, H 5.47, N 9.62.

*N*-(3-(1-hydroxyethyl)phenyl)-1H-indole-2-carboxamide (**S10**): A (0.76 mmol) of sodium borohydride (NaBH_4_) was added to a solution of *N*-(3-acetylphenyl)-1H-indole-2-carboxamide (**S9**) in 10 mL of THF. The reaction mixture was stirred overnight at room temperature, then the solvent was evaporated under reduced pressure and the desired precipitate was collected and dried.

White solid, yield 87%, m.p. 172–174 °C. ^1^H-NMR (500 MHz, CD_3_OD-*d4*) δ (ppm): 1.44 (d, *J* = 6.3 Hz, 3H); 3.27 (m, 1H); 5.81 (d, *J* = 7.5 Hz, 1H); 7.03–7.06 (t, *J* = 7.4 Hz, 1H); 7.13 (d, *J* = 7.5 Hz,1H); 7.20 (t, *J* = 7.6 Hz, 1H); 7.27–7.30 (m, 2H); 7.44 (d, *J* = 8.2 Hz, 1H); 7.59–7.61 (m, 2H); 7.70 (s, 1H); 9.93 (s, 1H, N-H amide); 11.08 (s, 1H, N-H indole). ^13^C-NMR (125 MHz, CD_3_OD-*d4*) δ (ppm): 24.2 (CH_3_); 69.4 (C-O); 103.8 (C8); 111.7 (C3); 117.8 (C2’); 119.4 (C6’); 119.9 (C6); 121.1 (C5); 121.5 (C7); 123.9 (C4’); 127.7 (C4); 128.4 (C5’); 131.0 (C1’); 137.2 (C2); 138.3 (C9); 147.1 (C3’); 160.9 (C = O amide). FT-IR (KBr-disc) cm^−1^: 3525 N-H amide; 3304 broad OH and N-H indole; 1647 C = O amide. Elemental analysis for C_17_H_16_N_2_O_2_: calcd. C 72.84, H 5.75, N 9.99; found C 72.53, H 6.08, N 9.54

### 3.3. Biological Methods

#### 3.3.1. Cell Maintenance and Cell Culture

Lung cancer cells were maintained in RPMI 1640 culture medium. The media were completed with 100 U/ mL pencillin and 0.1 mg/ mL streptomycin. Additionally, 10% FBS and 2 mM L-glutamine were added. Cells were cultured in 75 cm^2^ and incubated in incubator (5% CO_2,_ and 95% humidified air) at 37 °C.

Cells were cultured in RPMI 1640 media until they became confluent (90%), then the cultured media was removed. Flask was washed three times with 3 mL PBS, then the trypsin was added for 2–3 min in an incubator at 37 °C. When cells were de-attached from the flask (examined under microscope), media was added with continuous mixing to ensure single cell suspension. Then, cell suspension was transferred to 15 mL conical tube for counting. Hemocytometer was used to determine number of cells.

#### 3.3.2. Cell Viability Assay

In order to evaluate cell proliferation, the 3-(4,5-dimethylthiazol-2-yl)-2,5-diphenyltetrazoliumbromide (MTT) assay was employed as previously described [46]. A549 and normal dermal fibroblasts cells were plated in 96-well plates at a variable seeding density per well depending on doubling time, proliferation ability, and target time of treatment.

#### 3.3.3. Monoamine Oxidase A (MAO-A) Activity Assay

Before starting the MAO-A activity assay, lung cancer cells A549 were cultured in 6-well plates for 24 h, then treated with IC_50_ and sub-IC_50_ (a concentration that is lower than IC_50_) of MAO-A inhibitors for 48 h. A series of H_2_O_2_ standards (0, 200, 400, 600, 800, and 1000 pmol/well) was prepared by adding 0, 2, 4, 6, 8, and 10 µL of 0.1 mM H_2_O_2_. Further, the volume was adjusted to 50 µL/well with MAO assay buffer. Cells were dethatched using trypsin and centrifuged. One million cells were counted for each sample and homogenized using MAO assay buffer (0.1 mg/µL). The homogenates were centrifuged, and the supernatants were collected. Then, 10 µL of MAO-B inhibitor (Selegline) (Sigma Aldrich, Darmstadt, Germany) was added to each sample. Positive control was prepared by adding 4 µL of positive control solution and completing the volume to 50 µL with MAO assay buffer. Next, 50 µL of MAO reaction mix and background mix for each reaction was prepared. Both the reaction and background mix contained MAO assay buffer, developer, MAO substrate, and probe. Reaction and background mix were added into each standard, sample, and positive control. Finally, fluorescent measurement at (Ex/Em = 535/587 nm) was performed in kinetic mode at 25 °C for 60 min.

#### 3.3.4. Statistical Analysis

Data analysis was performed using GraphPad Prism software (GraphPad Prism version 8.0.0 for Windows, GraphPad Software, San Diego, CA, USA). The differences between treatment groups were determined by independent sample *t*-test. Paired sample *t*-test was used to assess cell viability over various time points. Data are expressed as mean ± SD, and *p* < 0.05 was considered a statistically significant difference. A non-linear regression analysis was used to calculate IC_50_ values.

## 4. Conclusions

The current study demonstrated the role of MAO-A enzyme in lung cancer as a promotor for proliferation, tumorigenesis, migration, and invasion of human lung cancer cells. We identified the pharmacophoric features of MAO-A inhibitors and disclosed potential hits targeting MAO-A binding domain. Moreover, we successfully conducted the design and synthesis of novel compounds that exerted their anticancer activities through MAO-A inhibition. Apparently, the ability of the synthesized inhibitors to affect the proliferation of cancer cells is not only mediated through their MAO-A inhibition but also crosslinks with several signaling pathways in the cancer cell, suggesting new roles of MAO-A at the molecular level. Herein, we provided the first evidence of MAO-A involvement in different lung cancer cell lines, and confirmed the ability of synthesized compounds to inhibit MAO-A selectively and suppress cancer growth.

## Figures and Tables

**Figure 1 molecules-27-02887-f001:**
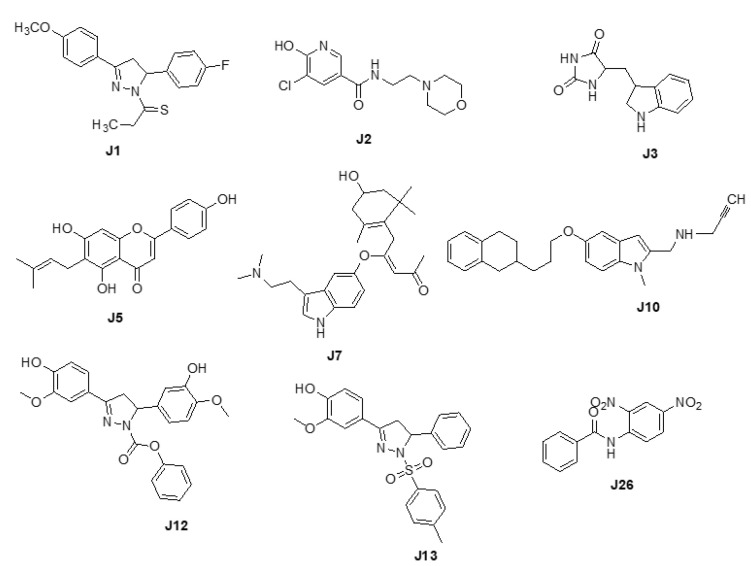
Chemical structures of **J1**-**J3**, **J5**, **J7**, **J10**, **J12**, **J13**, and **J26**.

**Figure 2 molecules-27-02887-f002:**
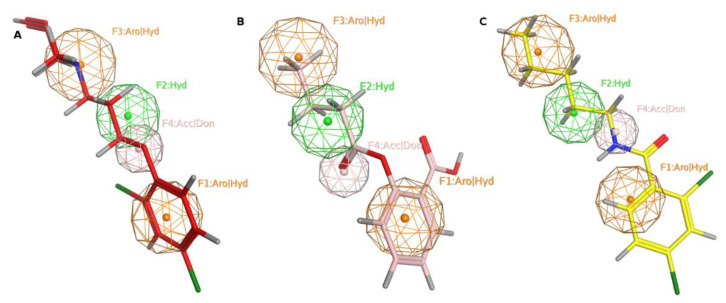
MAO-A pharmacophore model with (**A**) MLG (**S1**), (**B**) NSC215, and (**C**) NSC6884 Aro stands for aromatic ring, Hyd: hydrophobic, Acc: acceptor, Don: donor. Picture made by MOE [27].

**Figure 3 molecules-27-02887-f003:**
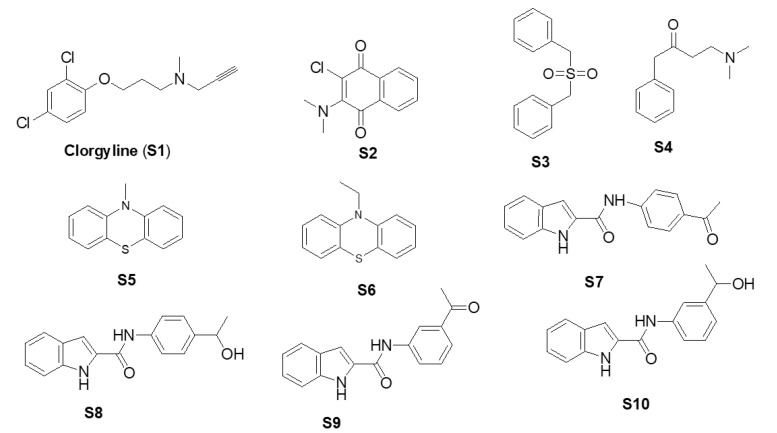
Chemical structures of **S1**, **S2–S6**, **S7**, and **S8–S10**.

**Figure 4 molecules-27-02887-f004:**
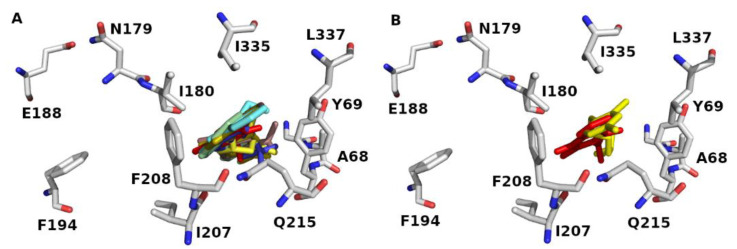
MAO-A (PDB ID: 2Z5X) binding domain (**A**) orientation of Glide docked poses of **J** series and (**B**) overlaying of the Glide docked pose of **J14** (red color) and original geometry of (clorgyline (**S1**) (yellow color). Some of key binding residues are shown, and H atoms are hidden for clarification. Picture visualized by PYMOL [32].

**Figure 5 molecules-27-02887-f005:**
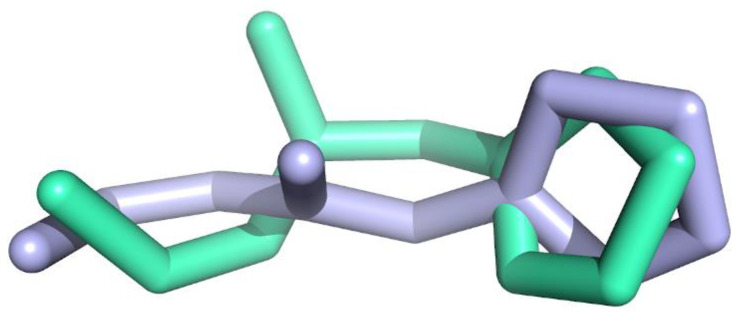
The superposition of the Glide-docked F2M pose (blue color) and its native conformation (green color) in 5MRL. The RMSD for heavy atoms of F2M is 1.048 Å. Picture made by PYMOL [32].

**Figure 6 molecules-27-02887-f006:**
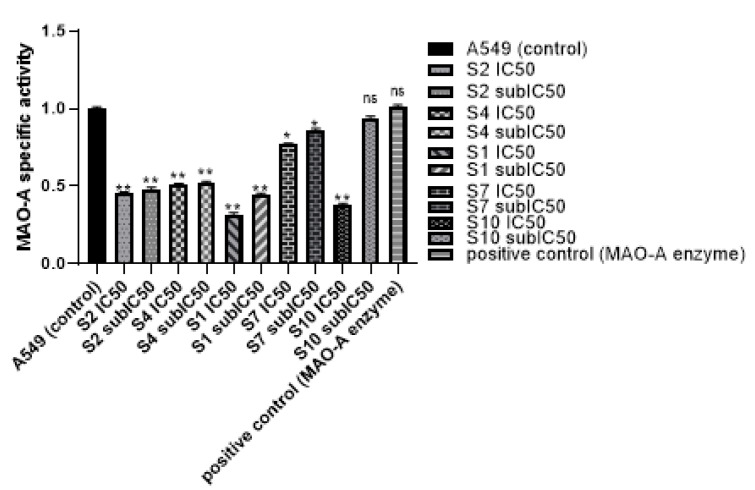
MAO-A activity assay. A549 cells were cultured and treated with IC_50_ and sub-IC_50_ of **S1**, **S2**, **S4**, **S7**, and **S10**. Fluorescence detection was on Ex/Em 535/587 nm. Data were expressed as mean + SD. SD did not exceed 5%. *p*-value < 0.05 indicates statistical significance in-comparison to untreated control, while asterisk: ns (not-significant) *p* > 0.05; * *p* ≤ 0.05; ** *p* ≤ 0.01 (according to GraphPad prism 9).

**Table 1 molecules-27-02887-t001:** The Glide scores (Kcal/mol) and H-bond against 2Z5X. NA stands for not available.

Compounds	Docking Score	Binding Residues	Compounds	Docking Score	Binding Residues
**S1**	−6.78	NA	**S6**	−7.82	NA
**S2**	−7.64	Asn181	**S7**	−9.59	Ala68, Asn181, Phe208
**S3**	−8.28	Ala68, Tyr69	**S8**	−9.78	Gly443
**S4**	−7.16	Ala68, Tyr69, Gln215	**S9**	−11.02	Ala68, Tyr69, Phe208
**S5**	−6.41	NA	**S10**	−10.26	Ala68, Tyr69, Phe208

**Table 2 molecules-27-02887-t002:** The 50% inhibitory concentration (IC_50_) values for MAO-A inhibitors in A549, H1299, and H661 lung cancer cells at 24, 48, 72 h. Experiments were carried out in duplicate at 4 independent times (*n* = 8). SD did not exceed 5%. SD: standard deviation, h: hour; µM: micromolar.

Cell Line	A549	H661	H1299	Cell Line	A549	H661	H1299
S1 IC_50_ (µM)				S6 IC_50_ (µM)			
24 h	367	252.3	330.4	24 h	>1000	>1000	>1000
48 h	208.99	215	323.6	48 h	>1000	>1000	>1000
72 h	198.2	240.5	188.5	72 h	>1000	>1000	>1000
S2 IC_50_ (µM)				S7 IC_50_ (µM)			
24 h	49.9	27.7	33.7	24 h	>1000	>1000	>1000
48 h	33.37	10.76	60.32	48 h	307.7	309.6	922.4
72 h	31.18	7.626	27.1	72 h	255.9	168.8	464.9
S3 IC_50_ (µM)				S8 IC_50_ (µM)			
24 h	>1000	>1000	>1000	24 h	>1000	>1000	>1000
48 h	>1000	>1000	>1000	48 h	>1000	875.3	>1000
72 h	>1000	>1000	>1000	72 h	>1000	163.8	>1000
S4 IC_50_ (µM)				S9 IC_50_ (µM)			
24 h	145.2	89.52	75.62	24 h	>1000	>1000	>1000
48 h	146.1	63.96	65.46	48 h	>1000	178.8	716.2
72 h	141	27.74	48.47	72 h	357	344.9	398.8
S5 IC_50_ (µM)				S10 IC_50_ (µM)			
24 h	>1000	>1000	>1000	24 h	>1000	>1000	>1000
48 h	>1000	>1000	>1000	48 h	147.2	73.83	116
72 h	>1000	>1000	>1000	72 h	143	44.79	139.8

**Table 3 molecules-27-02887-t003:** MAO-A activity assay. A549 cells were cultured and treated with IC_50_, sub-IC_50_ of **S1**, **S2**, **S4**, **S7**, and **S10**. Fluorescence detection was on Ex/Em 535/587 nm. Data were expressed as mean and SD did not exceed 5%. *p*-value < 0.05 indicates statistical significance in-comparison to untreated control, while asterisk: ns (not-significant) *p* > 0.05; * *p* ≤ 0.05; ** *p* ≤ 0.01; (according to GraphPad prism 9). µM: micromolar.

Sample Name	H_2_O_2_ (Pmol)	MAO-A Activity (Pmol/Min/ Ml)	Specific Activity Of MAO-A	*p*-Value
A549 (control)	111.14	18.52	1.00	-
**S2** IC_50_	50.19	8.36	0.45	0.0082 (**)
**S2** sub-IC_50_	53.37	8.89	0.48	0.0087 (**)
**S4** IC_50_	56.40	9.40	0.51	0.0091 (**)
**S4** sub-IC_50_	57.73	9.62	0.52	0.0094 (**)
**S1** IC_50_	35.3	5.883333333	0.317617477	0.0066 (**)
**S1** sub-IC_50_	49.35	8.225	0.444034631	0.0081 (**)
**S7** IC_50_	85.35	14.225	0.767950471	0.0193 (*)
**S7** sub-IC_50_	95.66	15.94333333	0.860716369	0.0321 (*)
**S10** IC_50_	42.04	7.006666667	0.37826172	0.0072 (**)
**S10** sub-IC_50_	104.47	17.41166667	0.939985773	0.0735 (ns)
positive control (MAO-A enzyme)	112.64	18.77333333	1.013496673	0.3171 (ns)

**Table 4 molecules-27-02887-t004:** Hits obtained by ligand- and structure-based drug design approaches. The chemical structure of hits was retrieved from NCI database. Hits have been provided by NCI agency.

NSC Code	Compound Code	Structure
-	Clorgyline (S1)	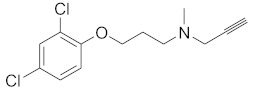
93074	S2	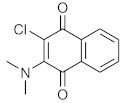
75	S3	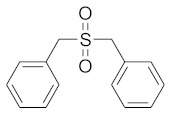
89702	S4	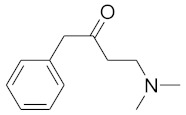
120	S5	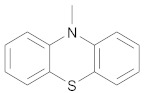
121	S6	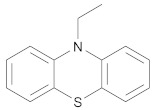

## Data Availability

Not applicable.

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
