# Peer review of "Design, Synthesis, and Biological Evaluation of Novel MAO-A Inhibitors Targeting Lung Cancer"

_molecules, 2022, doi:10.3390/molecules27092887_

Round 1
Reviewer 1 Report
The authors of the article present the design, synthesis and biological evaluation of new MAO-A targeting lung cancer inhibitors.
1. The introduction is correctly written, although it lacks the formulas of the sample compounds discussed in the introduction (pargyline, phenelzine, clorgyline, temzolomide, etc.). I would suggest including them as figure in the introduction.
2. The numbering J1-J3, J5, J7, J10, J12-J13, J19, J26 is not clear to me. What it results from, because it makes the reception of the article difficult. I suggest writing them down J1, J2, J3, J4 etc. Please add formula J19 to Figure 1.
3. Section 2.4 for rebuild. "Significant" is repeated too many times. Please change, rewrite your sentences.
4. All tables - please make them according to Molecules MDPI standards. Such bold lines are unacceptable.
5. Line 157 - there is no time in Supplementary Materials (SM) like "Figures 1-5", there are "Figures S1 - S5". Same line 163 - there is no "Table 1" in SM, but there is Table S1 ".
6. No gaps between individual sections.
7. Line 176 - lack of dashes, it should be "radio- and chemotherapies"
8. Line 191 should be "antiproliferative".
9. No explanation for "sub-IC50". I have not found it anywhere in the article.
10. Figure 6. - You cannot build a linear relationship with 2 measurements. How can you be sure that it is linear and not logarithmic? This is quite an abuse, unacceptable and I am asking for improvement.
11. Figure 6, Figure 7 - please remove figure cuts, the Molecules MDPI standard does not account for something like this.
12. There should be "mL" throughout the article, not "ml"
13. Section 3.2.2. - for complete reconstruction. No synthesis descriptions for compounds S2-S6. Please complete them. If the compounds S7-S10 are new, it is necessary to make and describe 1H, 13C, COSY, HSQC NMR, HR-ESI MS, FT-IR and optionally UV-VIS spectra. Additionally, please write down the form of the compounds, their color, and their melting/boiling points. In addition, please provide the purity of the compounds, if biological tests were performed and on what basis was this purity determined - NMR, GC, HPLC? Please include copies of all spectra in supplementary materials.
14. Section 3.2.2. - on what basis the authors claim that they obtained pure compounds without purifying them.
Author Response
Reviewer 1
The authors of the article present the design, synthesis and biological evaluation of new MAO-A targeting lung cancer inhibitors.
The introduction is correctly written, although it lacks the formulas of the sample compounds discussed in the introduction (pargyline, phenelzine, clorgyline, temzolomide, etc.). I would suggest including them as figure in the introduction.
Our response: We respect the reviewer’s suggestion to insert a new figure for the chemical structures of well-known drugs that were previously studied and reported. We think that an interested reader could retrieve structures information from search engine websites. Also, inserting this figure increases the number of figures and makes the manuscript bulkier.
The numbering J1-J3, J5, J7, J10, J12-J13, J19, J26 is not clear to me. What it results from, because it makes the reception of the article difficult. I suggest writing them down J1, J2, J3, J4 etc. Please add formula J19 to Figure 1.
Our response: We thank the reviewer for this comment. The numbering of J series was defined by Al-Janabi (co-author) and J letter was consequently derived. J19 is (Clorgyline: MLG: S1). Apologize for the mistake!
Section 2.4 for rebuild. "Significant" is repeated too many times. Please change, rewrite your sentences.
Our response: We thank the respected reviewer for the note. We did some rephrasing in the text to avoid the repetition of the word “significant”.
- All tables - please make them according to MoleculesMDPI standards. Such bold lines are unacceptable.
Our response: Tables were modified accordingly
- Line 157 - there is no time in Supplementary Materials (SM) like "Figures 1-5", there are "Figures S1 - S5". Same line 163 - there is no "Table 1" in SM, but there is Table S1 ".
Our response: Text was modified accordingly
No gaps between individual sections.
Our response: We thank the reviewer for this notice. We followed authors’ guidelines and fixed issues.
Line 176 - lack of dashes, it should be "radio- and chemotherapies"
Our response: Text was modified accordingly
Line 191 should be "antiproliferative".
Our response: Text was modified accordingly
No explanation for "sub-IC50". I have not found it anywhere in the article.
Our response:
We thank the respected reviewer for this valuable question.
Enzymatic activity was studied at 2 different concentrations, IC50 and sub-IC50 to rule out the possibility of enzyme degradation and loss of activity due to the killing effect of the inhibitors. In all cases, the specific activity of the enzyme was enhanced at sub-IC50 concentrations of the inhibitors, but the magnitude of this increase varied between the different inhibitors. We did not comment on that in the discussion part since this observation requires more in-depth study on the kinetics of the inhibition reaction which could be the scope of an upcoming paper.
- Figure 6. - You cannot build a linear relationship with 2 measurements. How can you be sure that it is linear and not logarithmic? This is quite an abuse, unacceptable and I am asking for improvement.
Our response:
We thank the respected reviewer for this note.
We apologize for this mistake. We never intended to show a relationship between the 2 measurements of the enzymatic activity at the 2 different concentrations since we did not study the kinetics of the inhibition reaction. We deleted this figure as it added nothing to Figure 7. We renamed Figure 7 to become Figure 6.
Figure 6, Figure 7 - please remove figure cuts, the MoleculesMDPI standard does not account for something like this.
Our response: We removed the frame.
There should be "mL" throughout the article, not "ml"
Our response: Text was modified accordingly
Section 3.2.2. - for complete reconstruction. No synthesis descriptions for compounds S2-S6. Please complete them. If the compounds S7-S10 are new, it is necessary to make and describe 1H, 13C, COSY, HSQC NMR, HR-ESI MS, FT-IR and optionally UV-VIS spectra. Additionally, please write down the form of the compounds, their color, and their melting/boiling points. In addition, please provide the purity of the compounds, if biological tests were performed and on what basis was this purity determined - NMR, GC, HPLC? Please include copies of all spectra in supplementary materials.
Our response: All relevant data of NMR and IR were provided in the Supplementary File accomplished with their spectra. The purity of prepared compounds have been examined using elemental analysis.
Section 3.2.2. - on what basis the authors claim that they obtained pure compounds without purifying them.
Our response: The purity of all prepared compounds have been examined using elemental analysis and the results lie in the accepted limits.
Reviewer 2 Report
Comments to the Author
The manuscript by Bardaweel et al. reported the synthesis and characterization of a series of small molecules applied structure-based anticancer activities through MAO-A inhibition. The detailed docking studies and MTT data reveal potential anticancer activities against several lung cancer cell lines. This present manuscript thus fits well into the Molecules readership's interests and may become suitable for publication after the revision. However, the issues need to resolve before the possible publication of these results.
- In the abstract section, an abbreviation such as MTT and IC50 needs to be defined in the first place in the manuscript before using their standard accepted abbreviations.
- The abstract section, "……prepared and characterized 1H-NMR, 13C-NMR, and IR." Both NMR and IR data are missing from the manuscript and supporting information file.
- In the introduction section (lines 49-50), the authors must provide a reference after the end of the sentence.
- For any potential drug candidate, purity is a crucial factor. Hence author should try for an HPLC study to confirm the purity of complexes.
- Line 171-172, relevant reference is required after the sentence's end.
- In the experimental section, the characterization data such as NMR and IR values should be given after the synthesis procedure of each compound.
Author Response
Reviewer 2
The manuscript by Bardaweel et al. reported the synthesis and characterization of a series of small molecules applied structure-based anticancer activities through MAO-A inhibition. The detailed docking studies and MTT data reveal potential anticancer activities against several lung cancer cell lines. This present manuscript thus fits well into the Molecules readership's interests and may become suitable for publication after the revision. However, the issues need to resolve before the possible publication of these results.
- In the abstract section, an abbreviation such as MTT and IC50 needs to be defined in the first place in the manuscript before using their standard accepted abbreviations.
Our response: Text was modified accordingly
- The abstract section, "……prepared and characterized 1H-NMR, 13C-NMR, and IR." Both NMR and IR data are missing from the manuscript and supporting information file.
Our response: All relevant data of NMR and IR were provided in the Supplementary File
- In the introduction section (lines 49-50), the authors must provide a reference after the end of the sentence.
Our response: A reference was inserted in the noted line.
- For any potential drug candidate, purity is a crucial factor. Hence author should try for an HPLC study to confirm the purity of complexes.
Our response: The purity of prepared compounds have been examined using elemental analysis.
- Line 171-172, relevant reference is required after the sentence's end.
Our response: A reference was inserted in the noted line.
- In the experimental section, the characterization data such as NMR and IR values should be given after the synthesis procedure of each compound.
Our response: All relevant data of NMR and IR are provided in the Supplementary File
Round 2
Reviewer 1 Report
Thank you for answering the questions. The article has been partially revised according to my recommendations. I also recommend that the article be read by a native speaker linguistically.
Things to improve:
1) The tables should be corrected in accordance with the standard contained in template Molecules MDPI: https://www.mdpi.com/files/word-templates/molecules-template.dot
2) Thank you for explaining what a sub-IC50 is. Please add an explanation of "sub-IC50" in the text of the article so that the reader, less familiar with the subject, also has an idea of what he is reading.
3) "Section 3.2.2. - on what basis the authors claim that they obtained pure compounds without purifying them.
Our response: The purity of all prepared compounds have been examined using elemental analysis and the results lie in the accepted limits. " However, it is not even mentioned in the text of the article - neither in methods, nor in biological research, nor in synthesis. Additionally, please explain what the "accepted limits" were and write them in the text of the article.
4) Compounds S1 - S6 were purchased or synthesized? Please write it down in the text of the article.
5) Elemental analysis, as well as NMR and IR measurements, were carried out on what equipment? Please include producers, cities and countries
6) The authors have still not provided the information requested: "Additionally, please write down the form of the compounds, their color, and their melting / boiling points." Please enter this information in the appropriate place in the article.
7) Thank you for placing the NMR and IR spectra in SM. In the text of the article, please include a list of all 1H- and 13C-NMR signals along with multiplicity, number of protons / carbons and couplings. For IR spectra, please list the main / characteristic bands in the text of the article. Standard as it is done in scientific articles.
Author Response
Comments and Suggestions for Authors
Thank you for answering the questions. The article has been partially revised according to my recommendations. I also recommend that the article be read by a native speaker linguistically.
Our response: We thank the respected reviewer for the time and effort to enhance our manuscript. The article was checked for language and corrections were made where appropriate.
Things to improve:
- The tables should be corrected in accordance with the standard contained in template Molecules MDPI: https://www.mdpi.com/files/word-templates/molecules-template.dot
Our response: We thank the respectful reviewer for this comment. Changes have been made and tables are blue highlighted.
- Thank you for explaining what a sub-IC50 is. Please add an explanation of "sub-IC50" in the text of the article so that the reader, less familiar with the subject, also has an idea of what he is reading.
Our response: We thank the respectful reviewer for this comment. Text was modified accordingly (P13, Line 379).
3) "Section 3.2.2. - on what basis the authors claim that they obtained pure compounds without purifying them.
The purity of all prepared compounds have been examined using elemental analysis and the results lie in the accepted limits. " However, it is not even mentioned in the text of the article - neither in methods, nor in biological research, nor in synthesis. Additionally, please explain what the "accepted limits" were and write them in the text of the article.
Our response: text was modified accordingly (P11, Line 298-307)
4) Compounds S1 - S6 were purchased or synthesized? Please write it down in the text of the article.
Our response: S1 and retrieved hits (S2-S10) were provided by NCI agency (Page 3, Line 93-94)
5) Elemental analysis, as well as NMR and IR measurements, were carried out on what equipment? Please include producers, cities and countries
Our response: text was modified accordingly (P11, Line 298-307)
6) The authors have still not provided the information requested: "Additionally, please write down the form of the compounds, their color, and their melting / boiling points." Please enter this information in the appropriate place in the article.
Our response: text was modified accordingly (P12).
7) Thank you for placing the NMR and IR spectra in SM. In the text of the article, please include a list of all 1H- and 13C-NMR signals along with multiplicity, number of protons / carbons and couplings. For IR spectra, please list the main / characteristic bands in the text of the article. Standard as it is done in scientific articles.
Our response: text was modified accordingly (P11,12,13).